# Analysis of Early Japanese Meteorological Data and Historical Weather Documents to Reconstruct the Winter Climate between the 1840s and the Early 1850s

5  Junpei Hirano[1], Takehiko Mikami[2], Masumi Zaiki[3]

[1]Faculty of Liberal Arts, Department of History, Teikyo University, 359 Otsuka Hachioji-Shi, Tokyo, 192-0395, Japan
[2] Faculty of Urban Environmental Sciences, Department of Geography, Tokyo Metropolitan University, 1-1 Minami-Osawa, Hachioji-Shi, Tokyo, 192-0397, Japan
10  [3] Faculty of Economics, Seikei University, 3-3-1 Kichijoji-Kitamachi, Musashino-shi, Tokyo, 180-8633, Japan

*Correspondence to: Junpei Hirano (jhirano06@gmail.com)*

**Abstract.** The East Asian winter monsoon causes orographic snowfall over the windward side of the Japanese islands (facing the Sea of Japan and the north-westerly winter monsoon flow) and negative temperature anomalies 15 around Japan. Daily weather information recorded in old Japanese diaries can provide useful information on the historical occurrences of snowfall days. Here, this information was combined with recently recovered early daily instrumental temperature data collected during the 19th century to reconstruct the occurrence of winter monsoon outbreak days (WMDs) from the 1840s to the early 1850s in Japan. Analyses of interannual and intra-seasonal variations in WMDs revealed active winter monsoon outbreaks in the early 1840s. In 1840/41 and 1841/42, these 20 synchronously occurred with extreme snow events reported in central and southern China. However, winter monsoon outbreaks were absent during the mid to late winters of the mid-1840s and 1853/1854. Freezing records of Lake Suwa in central Japan showed that it did not freeze during 1844/1845 and 1853/1854, which was in agreement with our finding of inactive winter monsoons in these years. Comparing the occurrences of WMDs with early instrumental surface pressure data revealed that WMDs were associated with the active phases of the 25 winter monsoon, as represented by an east-west surface pressure gradient over East Asia.

**1 Introduction**

Determining climate conditions before the 20th century is important for evaluating natural background climate variability because the anthropogenic effects on climate during this time were negligible. However, few meteorological data records are available from before the 1850s except for Europe and North America (Lamb, 1977). Historical climatologists have used documentary data as climate proxies to reconstruct past climate variations before the 19th century. Various kinds of documentary data, such as the timing of grain and wine harvests, plant phenology, the freezing of water bodies, and daily weather documents have been used to reconstruct past climate conditions; these approaches have been mainly applied in Europe (Brázdil et al., 2005; Labbé et al.,2019), China (Ge et al., 2016), and Japan (Mikami, 2008). These documentary data are particularly important because they deal with short-term climatic fluctuations from the most recent past (Bradley, 2014).

In Japan, daily weather information recorded in old diaries provides important climate information from the 18th and 19th centuries. Many weather diaries, kept in local government offices, large farmhouses, shrines, and temples, are now preserved in local libraries and museums (Mikami, 2008). Yoshimura (1993, 2007, 2013) compiled a Historical Weather Database (HWDB), based on information contained in these types of documents from various places. There are no official meteorological data from the Japan Meteorological Agency (JMA) prior to the construction of the Hakodate Meteorological Observatory in 1873; thus, historical daily weather documents are used to reconstruct climate variables before 1872.

Diary-based data from the windward side of Japan (facing the Sea of Japan and the northwesterly winter monsoon flow) are particularly important for reconstructing the wintertime climate. Outbreaks of cold air masses from the Eurasian continent are accompanied by snow clouds over the Sea of Japan, which causes heavy orographic snowfall on the windward side of Japan. Consequently, snowfall records in this area and negative temperature anomalies over Japan signify winter monsoon outbreaks. Several studies have attempted to detect historical outbreaks of the East Asian winter monsoon using snowfall records from historical weather documents (Fukaishi and Tagami, 1992; Hirano and Mikami, 2008; Mizukoshi, 1993). However, only a limited number of continuous diaries that record the weather on the windward side of Japan are available, so the sparse coverage of weather documents for this area causes uncertainty in reconstructed results. As snowfall is highly localized, it is difficult to distinguish a local snowfall event from one caused by the East Asian winter monsoon. To overcome these issues, it is necessary to use both historical weather documents and early instrumental temperature data to detect winter monsoon outbreaks. Previous studies have been hampered in this regard by a lack of digitized early instrumental data. However, several early instrumental temperature and pressure series from central and western Japan covering the period since 1819 have recently been recovered. These early instrumental temperature data are

of great value for detecting negative temperature anomalies over Japan caused by outbreaks of the East Asian winter monsoon.

Zaiki et al. (2006) used early instrumental temperature data to calculate a representative seasonal mean temperature series for western Japan reaching as far back as the 1820s. They suggested that a temporary warm epoch occurred around the 1850s. However, they did not analyze intra-seasonal variations of winter monsoon activity during this time. East Asian winter monsoon outbreaks occur on sub-monthly and sub-seasonal time scales (Abdillah et al., 2021). Therefore, to reconstruct winter climate patterns in detail, it is necessary to study winter

monsoon outbreaks using both interannual and intra-seasonal time scales.

    The occurrences of warm winters during the 1840s and 1850s have been suggested from the analyses of Lake Suwa, located in central Japan, freezing records (Fig. 1). Freeze-up dates of Lake Suwa have been shown to be positively correlated with early winter temperatures, so freeze-up dates have been used as a proxy when reconstructing early winter temperatures (Gray, 1974; Mikami and Ishiguro, 1998; Tanaka and Yoshino, 1982).

Lake Suwa did not freeze (i.e., open lake) in the winters of 1842/43, 1844/45, and 1853/54 (Fujiwara and Arakawa, 1954), suggesting that these winters were warm. However, there was much uncertainty around intra-seasonal variations in winter monsoon activity for these years.

    This study aimed to clarify the interannual and intra-seasonal variations of the East Asian winter monsoon for the period 1839/40–1853/54 using continuous historical daily documents that recorded the weather on the

windward side of Japan and early instrumental daily temperature data for Tokyo (Zaiki et al., 2006).

    The East Asian winter monsoon is one of the most active atmospheric circulation systems during the boreal winter (Miao et al., 2020). Consequently, the analysis of daily weather documents and daily temperature data is useful for interpreting not only synoptic weather patterns in Japan but also large-scale circulation patterns over East Asia. Moreover, knowledge of the activity of the East Asian winter monsoon over short time scales is

valuable for studies of the impact of climate on past societies.

## 2 Data

### 2.1 Historical daily weather records

Winter monsoon outbreak days (WMDs) were determined from two historical diaries. The two-volume Hirosaki

Clan Agency diary recorded daily weather and weather-related phenomena from 1661 to 1868 at a Hirosaki Domain local government office in northern Japan (Fig. 1; Fukuma, 2010, 2014). An example of a daily weather record indicates that the weather was "Cloudy, with snow falling from last night to this morning, accumulating about 10 cm. Occasionally snowfall today" (Fig. 2). Hirosaki, located in a typical Sea of Japan-side-type climate

zone, receives heavy orographic snowfall brought by winter monsoon (Suzuki, 1962). Takeda-Genemon's diaries contain almost continuous daily weather records from 1830 to 1980 recorded by several generations living in a large farmhouse in Kawanishi in northern Japan (Fig. 1). The weather records are presented in the municipal history of Kawanishi town as a weather diagram (History Compilation Committee of Kawanishi Town, 1979, 1983). According to Suzuki's (1962) climatic divisions, Kawanishi is also located in a typical Sea of Japan-side-type climate zone. The daily weather data from this diary covering the winter season (December 1 to February 28) for 1839/40–1853/54 from Yoshimura's (1993, 2007, 2013) HWDB were used.

## 2.2 Early instrumental data

Early instrumental temperature data were also used to detect WMDs. Daily morning temperature and pressure data observed at approximately 7:00 JST in Tokyo from 1838 to 1855 (the Reiken-koubo collection), as reported in previous studies (Zaiki et al., 2006), were used. These data were obtained from the Tokugawa government's bureau of Astronomy for Calendar Making of Edo (Tokyo). An example of temperature and pressure observations for the 17th and 18th of December 1838 recorded in the Calendar series (Reiken-koubo) is shown in Fig. 3. Temperature data during the winter season (December 1 to February 28) for the period 1839/40–1853/54 were used, except for the missing years of 1842/43 and 1843/44.

Large-scale circulation fields associated with the East Asian winter monsoon are characterized by east-west surface pressure gradients between the Siberian High and the Aleutian Low. The relationship between this pressure gradient over East Asia and WMDs was analyzed using early surface pressure observation data from Tokyo, Nagasaki, and Beijing (Fig. 4) recovered from previous studies (Können et al., 2003; Zaiki et al., 2006; Zaiki et al., 2008). Pressure data from Nagasaki were recorded at the Dutch settlement of Dejima in Nagasaki (Können et al., 2003). Data from the Nagasaki series from 1851/52 to 1853/54 were used. Pressure data in Tokyo are reported in the Reiken-koubo collection. Although pressure data from the Reiken-koubo collection are available for the period 1838–1855, there are several gaps and quality problems in this series during the 1840s (Zaiki et al., 2006). Therefore, surface pressure data from Tokyo were only used for the years 1850/51–1853/54. Surface pressure observations from Beijing were recently recovered for the period 1841–1855 (Zaiki et al., 2008), and Beijing data for the years 1850/51–1853/54 were used. Information about the type and accuracy of thermometers and barometers used during this period is very limited. The instruments were apparently brought from European countries as the data were recorded in western units, such as Fahrenheit and inch. The reliability of the original data was carefully inspected by comparing it with modern meteorological station data, and data that did not pass this inspection were eliminated.

**2.3 JMA weather and temperature data**

For the modern instrument period, weather data recorded at JMA observatories were used to identify WMDs. The two JMA observatories (Aomori and Yamagata) nearest to the locations of the diaries (Hirosaki and Kawanishi) were selected. The JMA recorded daily weather data during the daytime (06:00–18:00 JST) and nighttime (18:00-06:00 JST). It is thought that weather phenomena that occur during the night are ignored by observers in historical daily weather documents (Mikami, 1993). Therefore, we used only daytime weather observations to compare with those in the historical period. JMA weather data are available from the late 1960s to the present. However, the format of these weather descriptions at most observatories changed after the mid-1980s. The weather descriptions recorded at the JMA Aomori observatory for January 1980 and January 1989 were compared (Table 1). The weather description from 1989 is more detailed than that from 1980. After the mid-1980s, weather data from most of the JMA observatories were determined based on instrumental data (e.g., precipitation and cloud cover), which are not appropriate for comparisons with historical weather documents. For this reason, JMA weather data for the years 1968/69–1979/80 were used to detect WMDs for the modern instrumental period. In addition, daily minimum temperature data observed in Tokyo for the years 1968/69–1979/80 were also used for the detection of WMDs.

**2.4 Reanalysis data**

The U.S. National Oceanic and Atmospheric Administration, Cooperative Institute for Research in Environmental Sciences, and Department of Energy (NOAA-CIRES-DOE) Twentieth Century Reanalysis, Version 3 (20CRv3; Slivinski et al., 2019) was used to conduct a composite analysis of the circulation fields associated with WMDs. 20CRv3 covers the global atmosphere at a spatial resolution of 1.0 x 1.0° (latitude/longitude). Here, only surface pressure observations were assimilated as input data, and sea ice concentrations (HadISST2.3) and sea surface temperature fields (SODAsi.3 and HadISST2.2) were used as boundary conditions. Daily mean sea level pressure (SLP) data and 850 hPa-level temperature data for the period 1968/69–1979/80 were used to produce composite SLP and temperature field maps.

**2.5 Gridded daily precipitation data**

To produce composite maps of daily precipitation patterns associated with WMDs, high-resolution (0.05 x 0.05° latitude/longitude) daily precipitation gridded data for the Japanese islands (APHRO_JP V1207) were obtained from the Asian Precipitation - Highly Resolved Observational Data Integration Toward Evaluation water resources project (Kamiguchi et al., 2010).

**2.6 Station pressure data**

Surface pressure data from the JMA observatories at Nagasaki and Tokyo (Fig. 4) were used for the years 1968/69–1979/80. In addition, surface pressure data observed at Beijing International Airport obtained from the National Centers for Environmental Information (NCEI), NOAA, website were used.

**3 Methods**

**3.1 Definition of WMD**

First, daily weather documents from Hirosaki and Kawanishi were categorized into four types: snowfall, rain, fine, and cloudy, according to Yoshimura's (2013) methodology. When several different weather descriptions appeared

on the same day, weather categories were prioritized as follows: 1) snowfall, 2) rain, and 3) fine or cloudy (no precipitation). For example, when snowfall, rainfall, and cloudy conditions were described for one day, snowfall was adopted for the weather for that day. Using these categorized weather types, the weather criterion for a WMD was defined as follows: snowfall was recorded in both Hirosaki and Kawanishi. Next, the temperature criterion for a WMD was defined as follows: the daily temperature anomaly in Tokyo was negative. A WMD was

considered to have occurred when both the weather and the temperature met these criteria (see Fig. 5 for an example). The temperature criterion (i.e., negative temperature anomaly in Tokyo) effectively excluded local snowfall events in northern Japan (i.e., a local snowfall event that occurred only in northern Japan when the temperature elsewhere in Japan was warmer than the climatology). Finally, interannual and intra-seasonal variations in WMDs were analyzed for the years 1839/40–1853/54. In addition, associations between the WMDs

and intra-seasonal variations of the east-west surface pressure gradient over East Asia (ΔSLP) were investigated using early surface pressure data from Tokyo, Nagasaki, and Beijing. The surface pressure differences between Beijing and Tokyo (ΔSLP B-T) and between Nagasaki and Tokyo (ΔSLP N-T) were calculated.

**3.2 Analysis of circulation and precipitation patterns associated with WMDs**

A lag composite analysis of the circulation fields using 20CRv3 was conducted to clarify the characteristics of circulation fields associated with WMDs. For this purpose, the modern occurrence dates of WMDs (1968/69–1979/80) were identified from JMA weather data using the same methodology used for the 19th-century data. The lag composite of the daily mean SLP and 850 hPa-level temperature were then calculated using 20CRv3. The WMDs and the preceding four non-WMDs were selected for the lag composite analysis. If WMDs continuously

appeared for several days, the first WMD and the preceding four non-WMD days were used. As a result, 55

WMDs were selected for the composite analysis. Lag composite analysis was also conducted for the daily precipitation patterns using the APHRO_JP gridded precipitation data.

To determine whether weather and temperature data could capture the East Asian winter monsoon activity on an intra-seasonal time scale, case studies were conducted for intra-seasonal variations of ΔSLP and WMDs. The coldest and warmest years were selected based on Japanese modern winter (i.e., December, January, February) mean temperatures from 1968/69 to 1979/80. A case study was then conducted for the selected winters.

## 4 Results

### 4.1 Circulation and precipitation patterns associated with WMDs

The temporal evolution of the SLP fields associated with WMDs is shown in Fig. 6. The anticyclonic anomaly over central Siberia on day-3, representing the Siberian High, gradually expanded southwards from day-2 to day-0, see Fig. 6(b)–(e). The cyclonic anomaly over eastern China moved northeastward from day-3 to day-2, see Fig. 6(b)–(c), implying that the extratropical cyclone migrated along the south-coast of Japan ("south-coast cyclone"; Ando and Ueno, 2015; Tasaka, 1980; Ueno, 1993; Yamazaki et al., 2015). The south-coast cyclone was amplified when it reached the Northern Pacific. A strong east-west SLP gradient appeared over Japan from day-1 to day-0, see Fig. 6(d)–(e), representing an active phase of the East Asian winter monsoon.

Temporal variations of 850 hPa-level temperature anomaly patterns are presented in Fig. 7. Dipole-like warm and cold anomalies occurred over eastern Eurasia on day-3, see Fig. 7 (b). Cold anomalies over Siberia gradually intensified over East Asia and extended southwards as far as Taiwan on day-0, see Fig. 7(e), implying that there was an outbreak of the cold air mass over East Asia.

Figure 8 presents the lag composite of daily precipitation patterns from day-4 to day-0. The precipitation area spread over both the Pacific Ocean and Sea of Japan sides of Japan from day-4 to day-1, see Fig. 8(a)–(d). Precipitation over the Pacific Ocean side was caused by the passage of the south-coast cyclone (Tasaka, 1980, 1988; Ueno, 1993). Meanwhile, the precipitation area on day-0 was limited to the Sea of Japan side, see Fig. 8(e), implying that orographic precipitation (snowfall) was brought by the northwesterly winter monsoon.

These composite analyses showed typical circulation and precipitation patterns associated with the East Asian winter monsoon outbreak (Abdillah et al., 2021), suggesting that it is reasonable to reconstruct WMDs in historical periods using weather and temperature data recorded in Japan.

Intra-seasonal variations in ΔSLP B-T and WMDs for a cold winter year (1976/77) and those for a warm winter year (1978/79) are presented in Fig. 9. WMDs were interpreted to occur more frequently in 1976/77 than in 1978/79. The peaks of ΔSLP in both years showed relatively good agreement with the WMDs. Similar results

were obtained from the analysis based on ΔSLP N-T and WMDs (Fig. 10), suggesting that it was reasonable to use WMDs as an indicator of intra-seasonal variations in East Asian winter monsoon activity.

**4.2 Reconstructed WMDs from the 1840s to the early 1850s**

Interannual and intra-seasonal variations in WMDs from the 1840s to the early 1850s are discussed in this section; interannual variations in the frequency of WMDs are presented in Fig. 11, and the WMDs for each year are presented in Fig. 12.

Three years, 1839/40, 1840/41, and 1841/42, were characterized by frequent WMDs. Kusakabe (1978) reported heavy snowfall on the windward side (Sea of Japan side) of central Japan in 1839/40 based on climate disaster chronology. Heavy snowfall and extremely cold weather in central Japan were also reported in 1840/41 by Kusakabe (1978). These records agree with the strong winter monsoon activities predicted here for both years. The occurrence of WMDs was low from mid to late winter during the mid-1840s. Only three WMDs occurred in 1844/45. It should be noted that Lake Suwa in central Japan did not freeze (i.e., open lake) in 1844/45 (Fujiwara and Arakawa, 1954), suggesting that this winter was warm. An absence of WMDs in mid to late winter was also observed in 1845/46, 1846/47, and 1847/48. Kusakabe (1978) reported that the winter of 1846/47 was extremely warm in central and western Japan: "there were a few falls of snows in this winter". These records agree with an inactive winter monsoon outbreak.

WMDs frequently occurred in mid to late winter in the 1850s, unlike the mid-1840s. However, it is noteworthy that the seasonal pattern of WMDs in 1853/54 was quite unusual. WMDs frequently occurred in December, but they did not occur after early January. This inactive winter monsoon outbreak after early January seemed to be associated with the Lake Suwa records, which showed that there was no freezing (Fujiwara and Arakawa, 1954).

Comparisons of temporal variations in ΔSLP and the occurrence of WMDs are shown in Fig. 13 (ΔSLP B-T) and Fig. 14 (ΔSLP N-T). SLP data from Nagasaki were not available for 1850/51. Thus, ΔSLP N-T is presented for 1851/52, 1852/53, and 1853/54. The WMDs were associated with the active phase of the East Asian winter monsoon (represented by ΔSLP). For example, WMDs occurred during almost all the peaks of ΔSLP B-T in 1852/53, see Fig. 13(c). Good agreement was also observed for 1853/54, see Fig. 13(d). A large-amplitude ΔSLP B-T was observed in early winter (December) associated with frequent WMDs, while the weak amplitude (weak intra-seasonal variations) of ΔSLP B-T after mid-January corresponded to the absence of WMDs. Thus, the unusual seasonal pattern of the winter monsoon in 1853/54 was also confirmed by the ΔSLP.

**5 Discussion and Conclusions**

Historical daily weather documents are useful proxy climate data for reconstructing past climates on daily to sub-seasonal time scales. However, weather records from diaries are qualitative and subjective. The locations of continuous diaries are unevenly distributed, making spatial analysis difficult. Therefore, when reconstructing past climates on short time scales, it is desirable to use both historical daily weather documents and early instrumental meteorological data.

This study reconstructed East Asian winter monsoonal activity around Japan from the 1840s to the early 1850s using historical daily weather documents and early instrumental temperature data. The results showed active winter monsoon outbreaks in the early 1840s, and inactive winter monsoon outbreaks occurring during the mid to late winters of the mid-1840s. Similar inactive winter monsoon activity was also observed in 1853/54. Records from Lake Suwa showed that it did not freeze 1844/45 or 1853/54, which was in good agreement with these inactive winter monsoon outbreaks. Comparing WMDs and ΔSLP confirmed that the timing of the WMDs was closely associated with the active phase of the East Asian winter monsoon.

Studies on modern weather periods have clarified that there are several different cold surge pathways over East Asia (Abdillah et al., 2021) that are controlled by large-scale atmospheric circulations (e.g., Siberian High expansion over the Eurasian continent). To understand the spatial extent of anomalous winter monsoon circulation, a regional comparison of cold surge activities over East Asia is of great value. We compared our results with extreme snow events over central and southern China reconstructed from Chinese historical documents (Hao et al., 2011).

In the current study, active winter monsoon outbreaks were detected in 1839/40, 1840/41, and 1842/42. Except for 1839/40, central and southern China also experienced extreme snow events (see Hao et al., 2011, p.165). Snow events in 1840/41 were characterized by heavy snow over scattered regions of eastern China, with cold dry conditions and a period of heavy snowfall starting on December 18 and continuing until January 2. Snowfall events in 1841/42 were characterized by persistent snowfall between 26°N–35°N in eastern China, with a probability of occurring once every 100 years. This snow event in 1841/42 started on December 9 and lasted until January 10. Heavy snowfall periods in both years were consistent with periods of frequent WMDs in Japan (Fig. 12). This suggested that anomalous cold surges synchronously affected both Japan and China during these winters.

No extreme snow events were reported in central and southern China from the mid-1840s to the early 1850s. This absence of extreme snowfall events in China seems to be in accordance with inactive winter monsoon outbreaks in Japan. Further studies are needed to clarify the mechanism of this coherence of cold surge activities in East Asia. Instrumental data from the mid-to high-latitude areas of the Eurasian continent would be of great

value for this purpose. Meteorological tables in *Annuaire magnétique et métérologique du Corps des ingénieurs des mines de Russie* and *Annales de l'observatorie physique central de Russie* (Kupffer, 1850, 1851, 1852, 1853a, 1853b, 1855a, 1855b, 1856, 1857) reported meteorological observations from many locations in Russia from 1841 to 1860. We plan to digitize all the surface pressure data reported in these books to help analyze the impact of the Siberian High and Arctic Oscillation on the East Asian winter climate on both interannual and intra-seasonal time scales.

The El Niño–Southern Oscillation (ENSO) is also an important factor that affects the East Asian winter monsoon. Typically, El Niño is associated with warm winters in Japan (Halpert and Ropelewski,1992). However, a recent study revealed that some El Niño events do not correspond to warm winters in East Asia (Shiozaki et al., 2021). Therefore, a direct comparison between historical El Niño chronology (Ortlieb, 2000; Quinn and Neal, 1992) and reconstructed East Asian winter monsoon outbreaks is inappropriate. Studies on modern weather periods have revealed that ENSO-related teleconnection patterns triggered by tropical forcing are the key system that modulates the East Asian winter monsoon (Sakai and Kawamura, 2009; Ueda et al., 2015; Wang et al., 2000). Further data rescue activities and climatic reconstructions of tropical and subtropical areas are needed to better understand ENSO-related teleconnection patterns that have affected East Asian winter monsoons in historical periods.

**Data availability**

All the data used to perform the analyses in this study have been described and properly referenced in this paper. Historical daily weather documents in the Historical Weather Database and surface air pressure observations in Tokyo and Nagasaki are available from the JAPAN-ASIA CLIMATE DATA PROGRAM (https://jcdp.jp) website. Surface air pressure observations in Beijing are reported in *Annuaire magnétique et métérologique du Corps des ingénieurs des mines de Russie* and *Annales de l'observatorie physique central de Russie*. These observations have been imaged through the National Oceanic and Atmospheric Administration (NOAA) Central Library Climate Data Imaging Project. Most modern meteorological data in Japan are available from the Japan Meteorological Agency (https://www.data.jma.go.jp/obd/stats/data/en/smp/index.html). Historical high-resolution daily precipitation gridded data over the Japan islands (APHRO_JP V1207) are available from the Asian Precipitation Highly Resolved Observational Data Integration Toward Evaluation of Water Resources project (http://aphrodite.st.hirosaki475 u.ac.jp/index.html) website.

**Abbreviation list**

JMA: Japan Meteorological Agency

WMD: Winter Monsoon Outbreak Day

HWDB: Historical Weather Database

20CRv3: Twentieth Century Reanalysis, Version 3

ΔSLP B-T: Surface Pressure Differences between Beijing and Tokyo

ΔSLP N-T: Surface Pressure Differences between Nagasaki and Tokyo


**Author contributions**

J. H. collected the data and performed most of the analyses with the guidance of T. M., who designed the research method, supervised the study, and assisted with interpreting the results. M. Z. collected and analyzed the early surface pressure observation data in Japan and China. J. H. and T. M. drafted the Figs and wrote the text. All the

authors participated in the analyses and helped to improve the article.

**Competing interests**

The authors declare that they have no conflict of interest.

**Acknowledgements**

We would like to thank Dr. Minoru Yoshimura (Emeritus Professor at Yamanashi University) for providing historical weather records compiled in the Historical Weather Database.

**Financial support**

This study was supported by JSPS KAKENHI Grant 20H01389, 19K01163, and 18H03794 from the Japanese Ministry of Education Science, Sports and Culture.

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

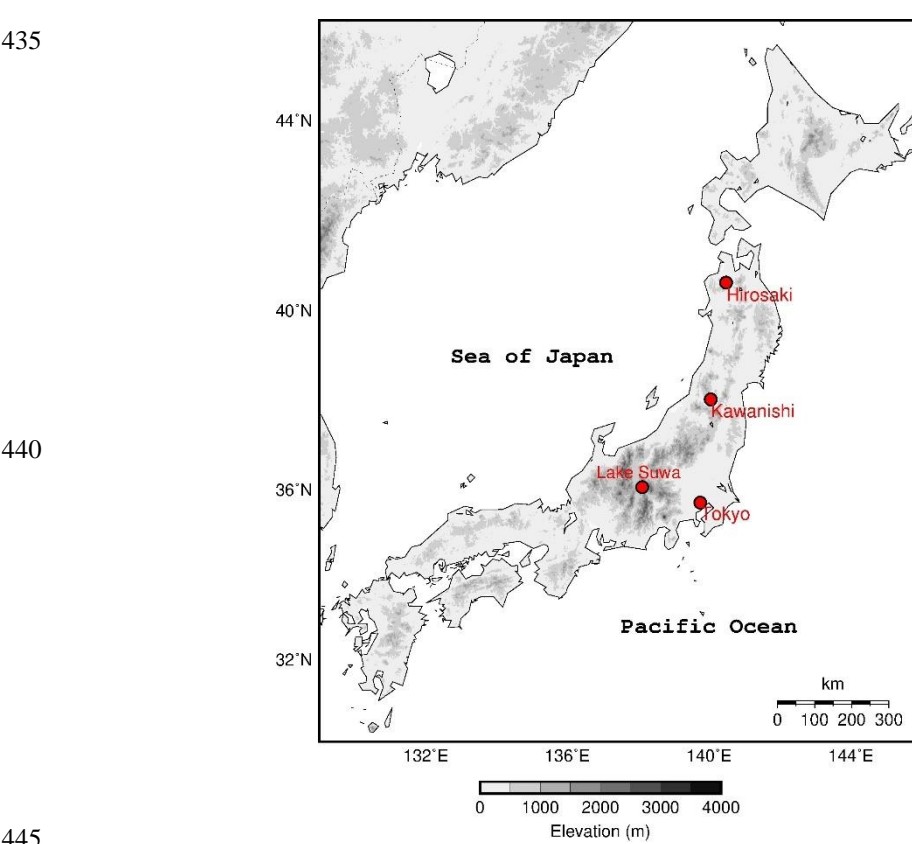



**Fig. 1 Locations of the historical daily weather records, Lake Suwa, and the instrumental temperature data used in this study.**


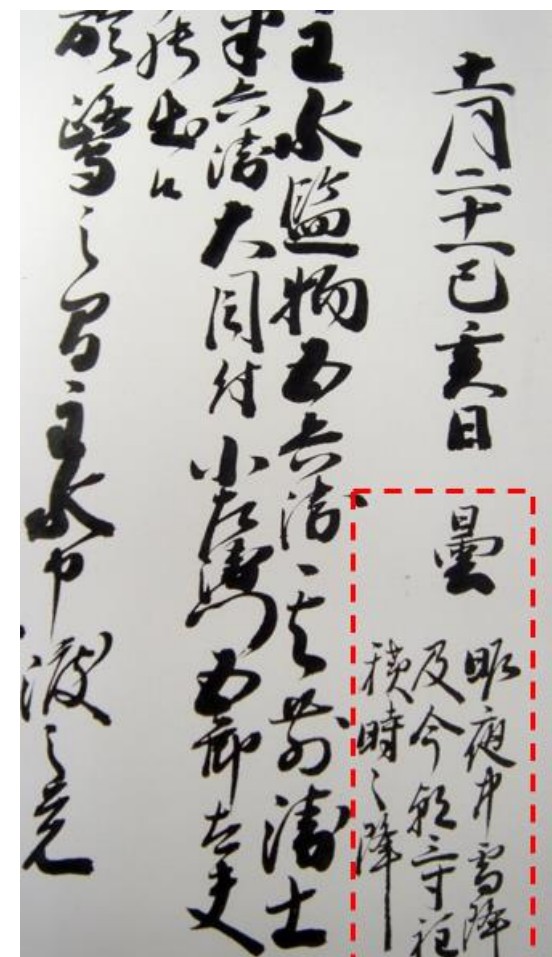




**Fig. 2 An example of daily weather records from the diary of the Hirosaki clan (for January 5, 1801) from the Hirosaki City Library collection. The weather information is shown in the area surrounded by red dashed lines.**

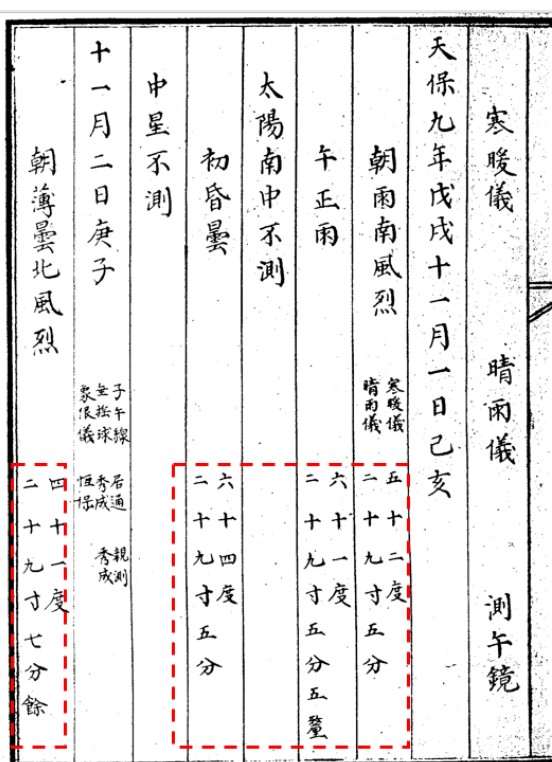

**Fig. 3 Record for the 17th and 18th of December 1838 from the Calendar series (Reiken-koubo) stored in the National Archives of Japan. This is the first page of the temperature and pressure observations, which are shown in the area surrounded by red dashed lines.**

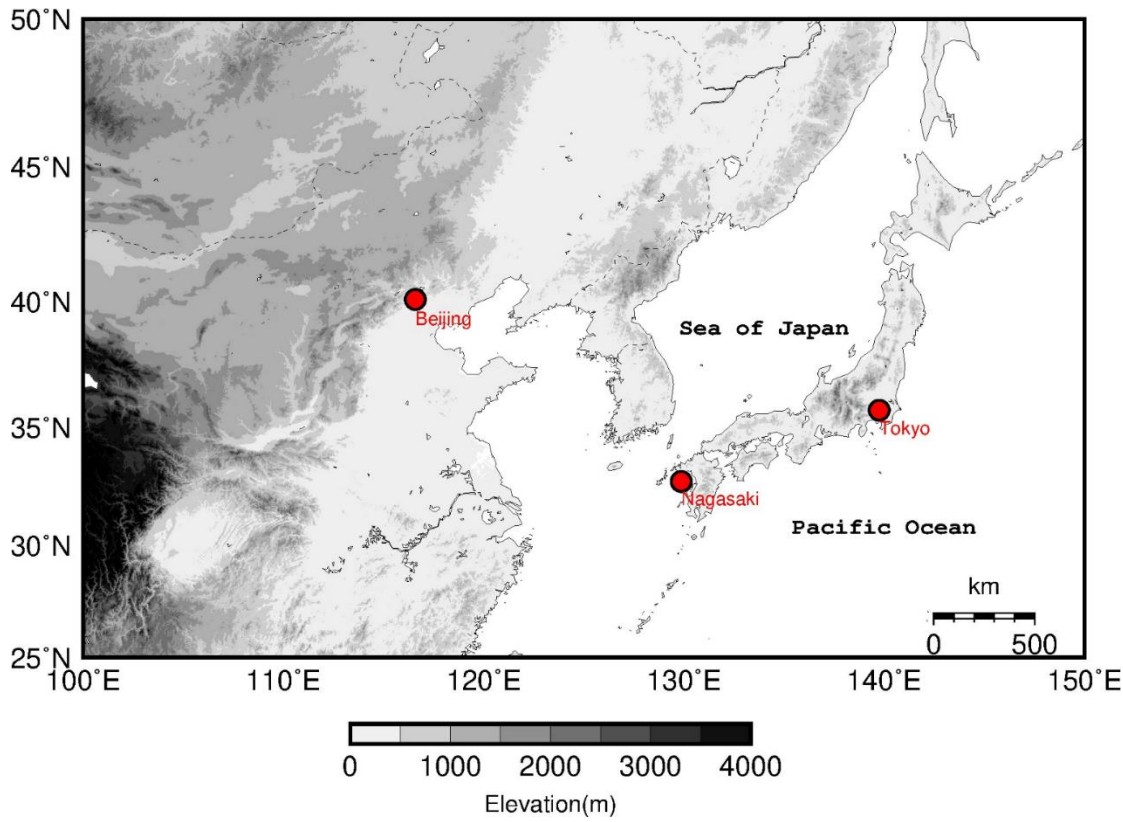

**Fig. 4 Locations of the early instrumental surface pressure data records from JMA observatories used in this study.**


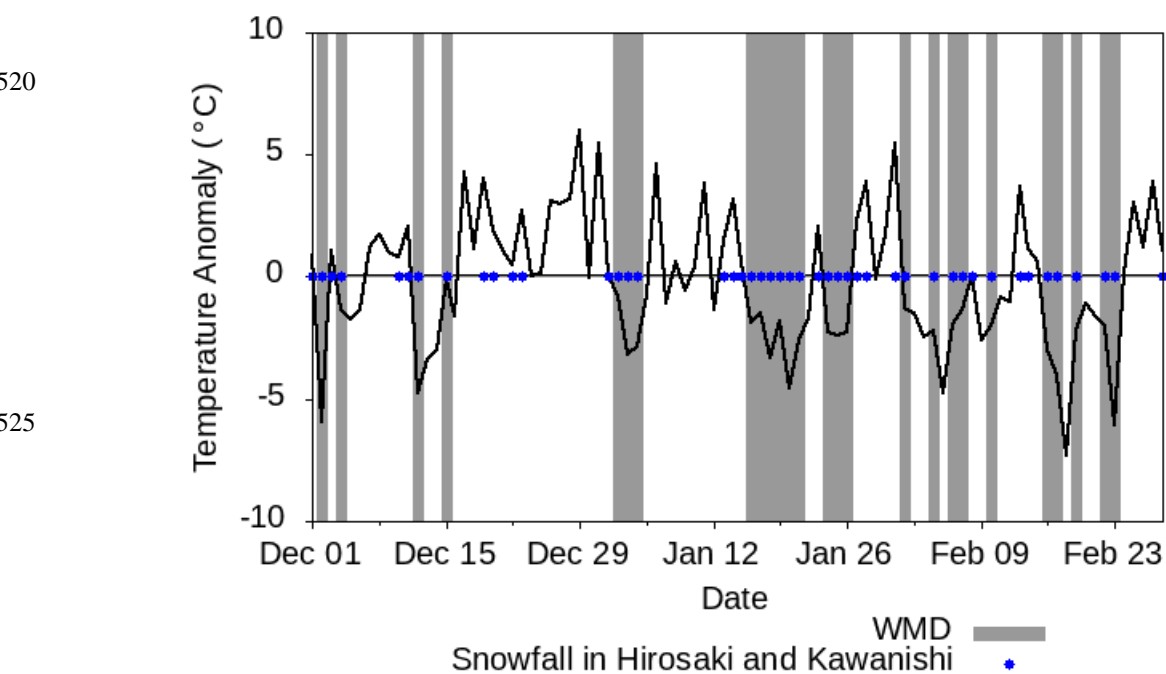

**Fig. 5 Time series of daily temperature anomalies in Tokyo (black solid line) and the dates of winter monsoon outbreak days (WMDs) for 1851/52. The blue dots indicate snowfall in Hirosaki and Kawanishi, and the gray shaded bars indicate WMDs. Temperature anomalies were calculated as deviations from daily climatology 1839/40–1853/54.**






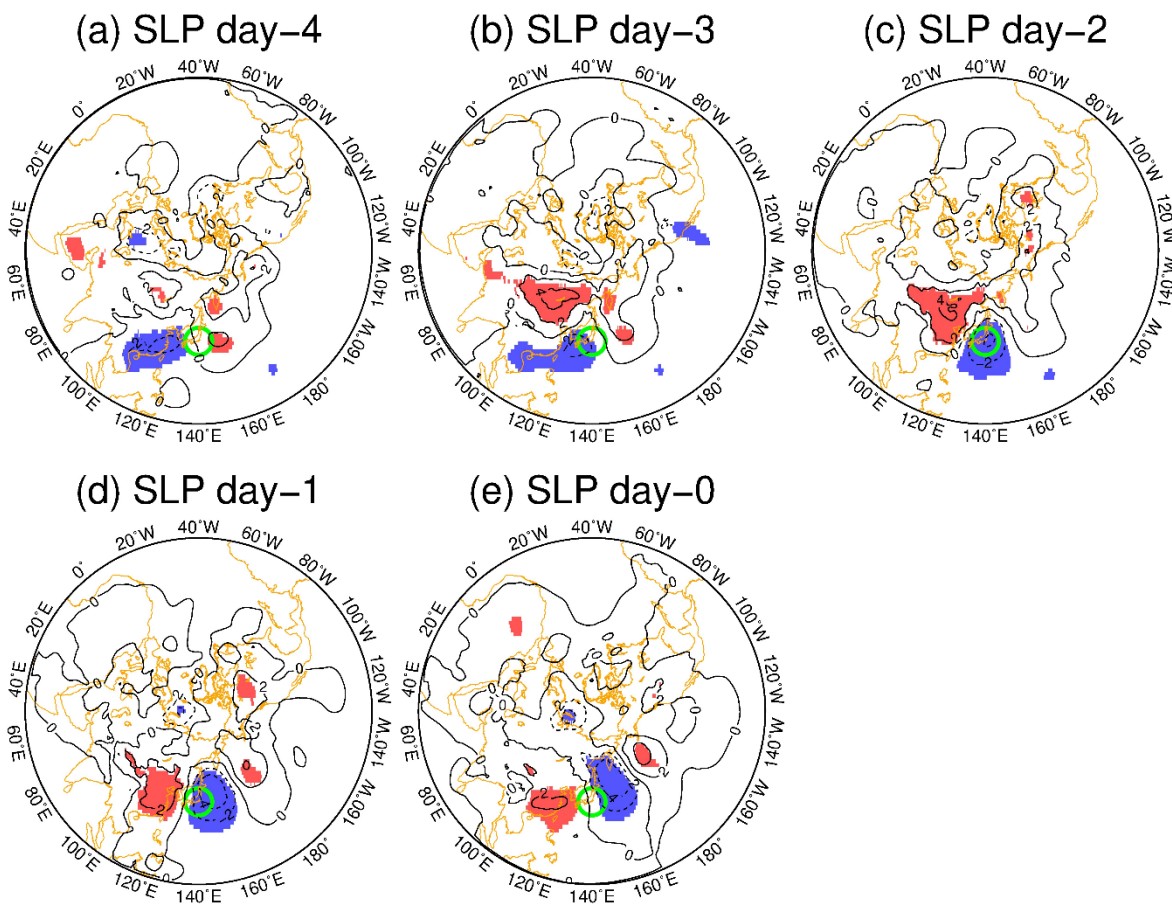


**Fig. 6 Composite daily mean sea level pressure (SLP; hPa) from day-4 to day-0 for 1968/69–1979/80. The contour interval is 2 hPa, and the red and blue shading denotes positive and negative anomalies significant at the 95 % confidence level, respectively, based on a two-tailed Student's t-test. A green circle indicates the position of Japan.**



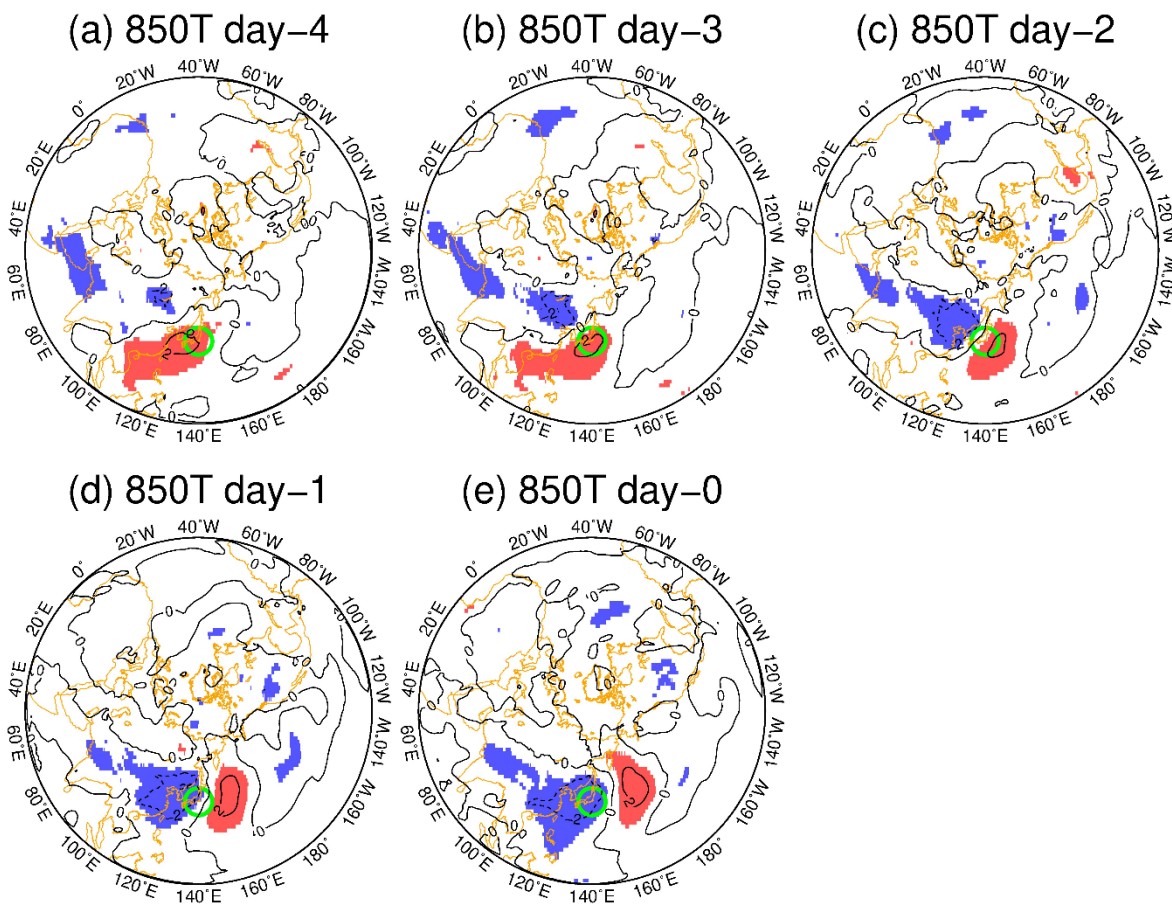

**Fig. 7 Composite daily mean 850 hPa temperature (°C) from day-4 to day-0 for 1968/69–1979/80. The contour interval is 2°C, and the red and blue shading denotes positive and negative anomalies significant at the 95 % confidence level, respectively, based on a two-tailed Student's t-test. A green circle indicates the position of Japan.**



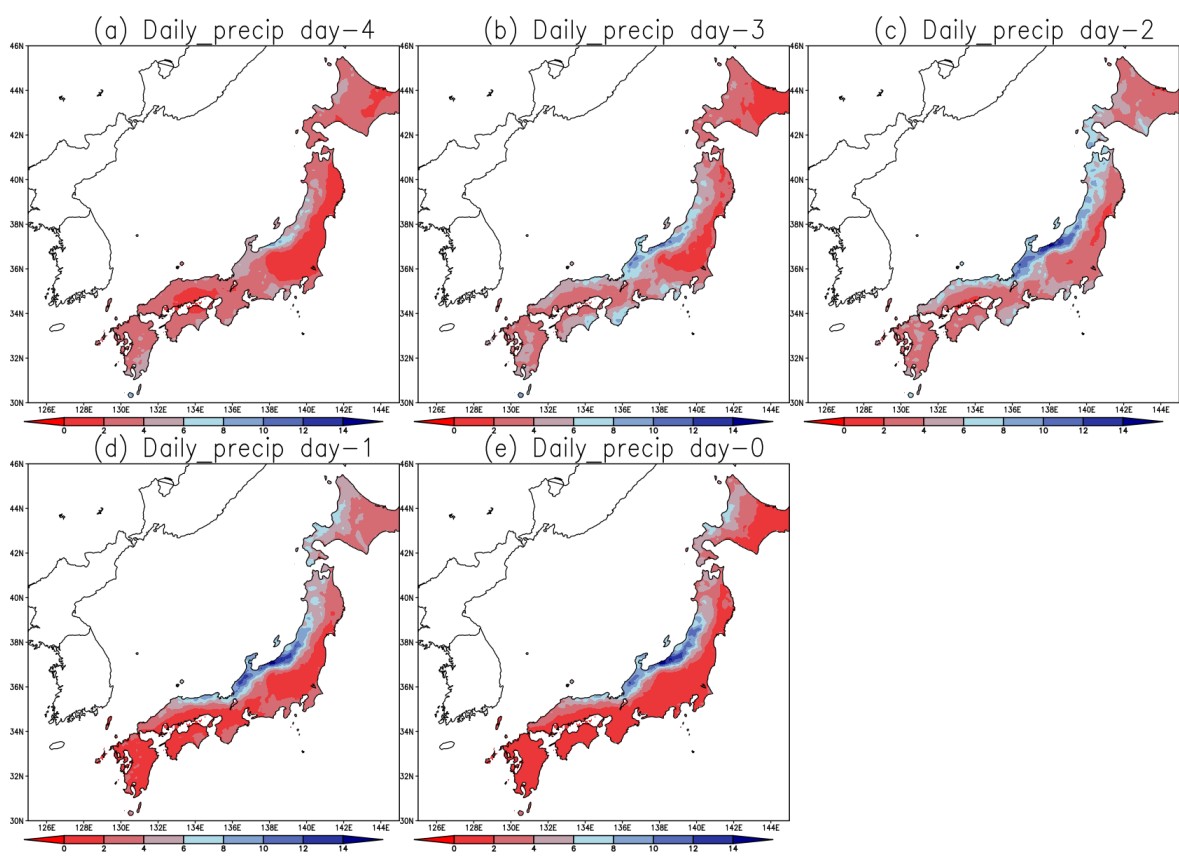


**Fig. 8 Composite daily mean precipitation (mm·day$^{-1}$) from day-4 to day-0 for 1968/69–1979/80.**




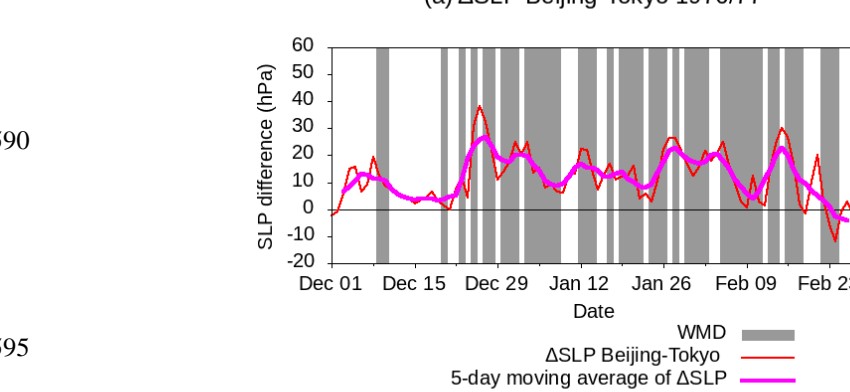



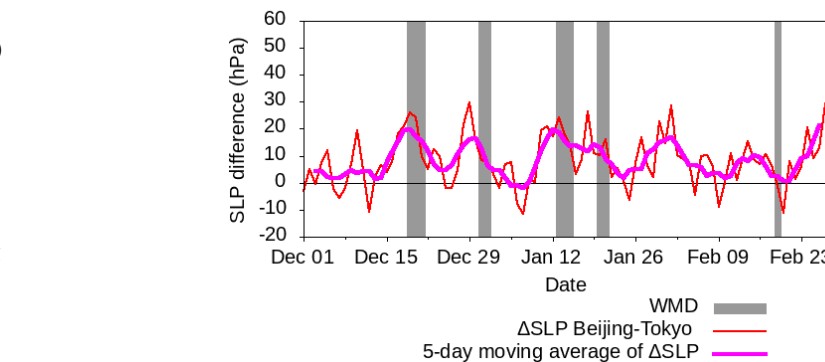



**Fig. 9 Temporal variations in intra-seasonal variations of east-west surface pressure differences between Beijing and**
**Tokyo (ΔSLP B-T) and the dates of winter monsoon outbreak days (WMDs). (a) Cold winter year (1976/77) and (b)**
    **warm winter year (1978/79).**



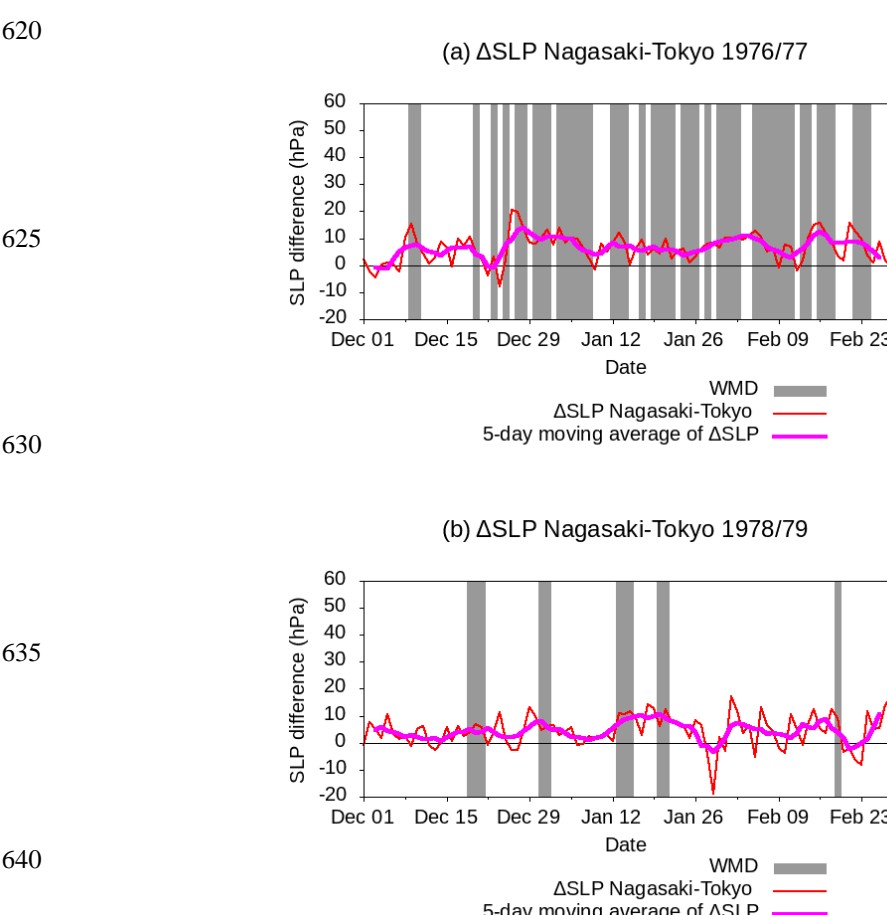




**Fig. 10 Temporal variations in intra-seasonal variations of east-west surface pressure differences between Nagasaki and Tokyo (ΔSLP N-T) and the dates of winter monsoon outbreak days (WMDs). (a) Cold winter year (1976/77) and (b) warm winter year (1978/79).**


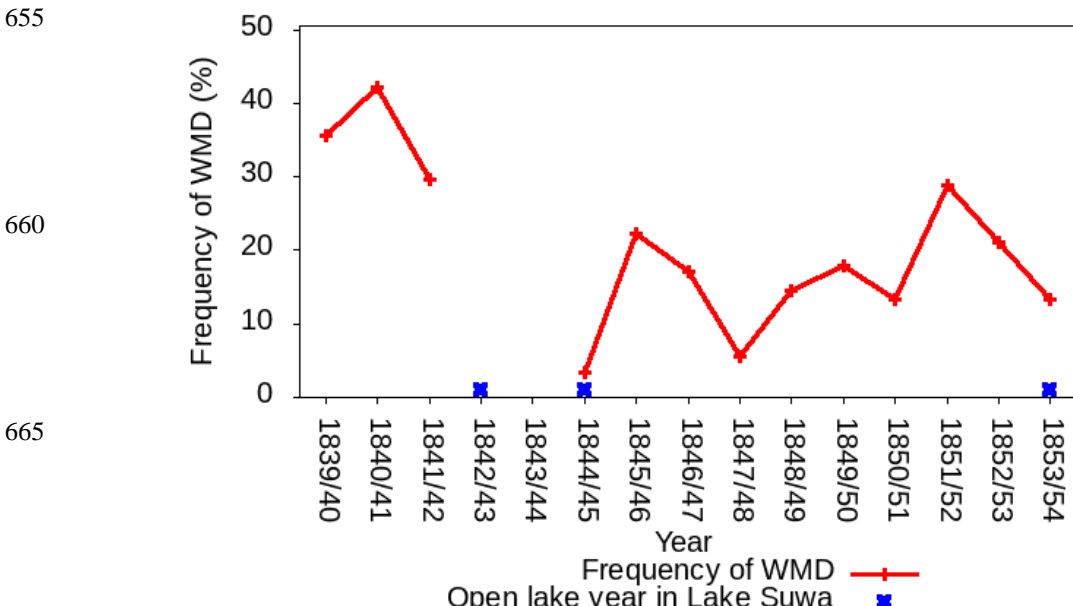

**Fig. 11 Interannual variations in the frequency of winter monsoon outbreak days (WMDs) for the years 1839/40–1853/54. The red solid line indicates the frequency of WMDs, and the blue dots indicate Lake Suwa's open-lake years.**

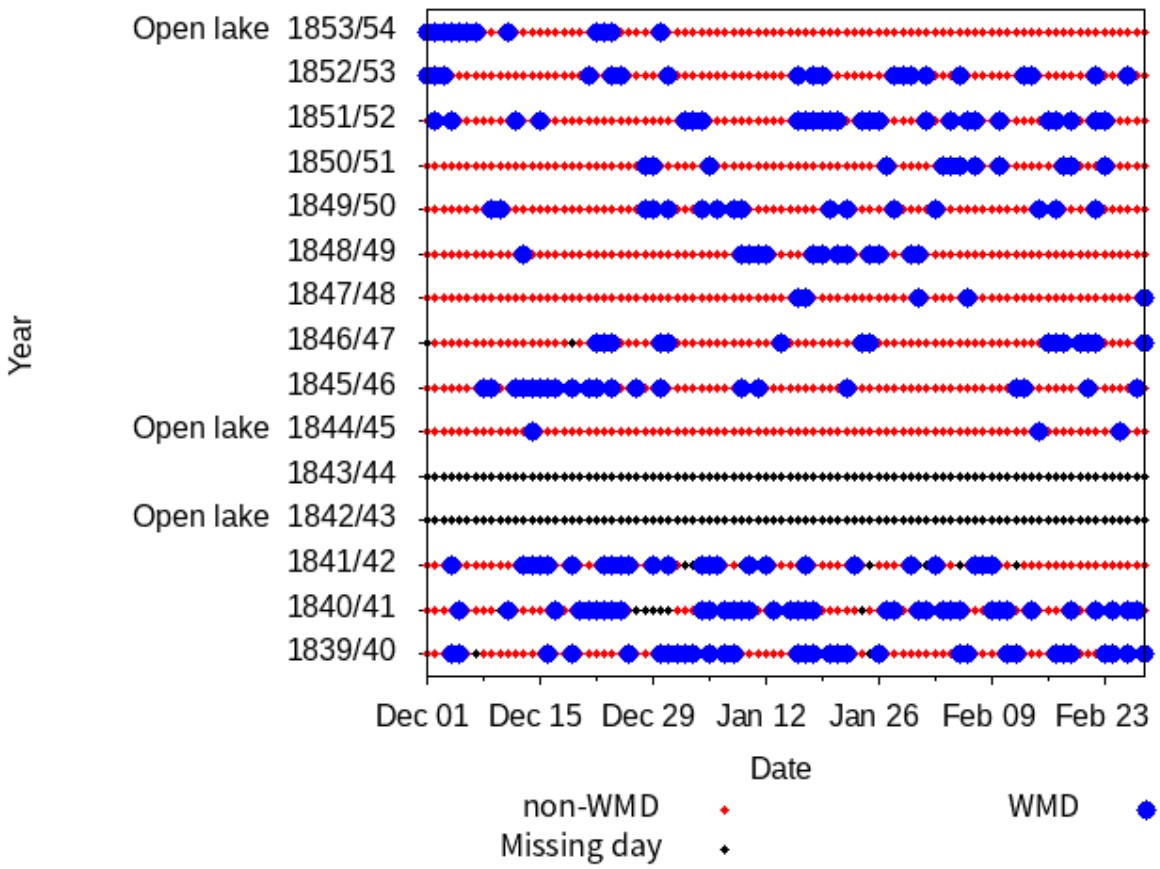

**Fig. 12 Winter monsoon outbreak days (WMDs) for the years 1839/40–1853/54. The blue dots indicate WMDs, the red dots indicate non-WMD days, and the black dots indicate days with no data.**

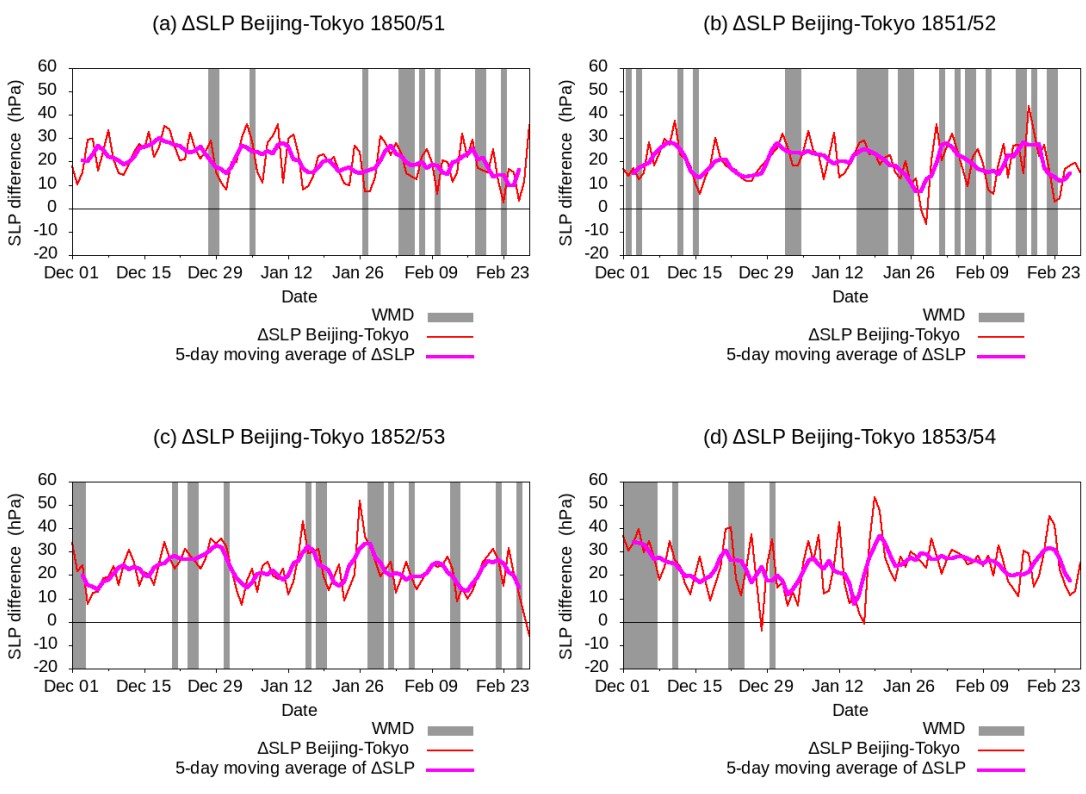


**Fig. 13 Temporal variations in surface pressure differences between Beijing and Tokyo (ΔSLP B-T) and winter monsoon outbreak days (WMDs) for (a) 1850/51, (b) 1851/52, (c) 1852/53, and (d) 1853/54.**



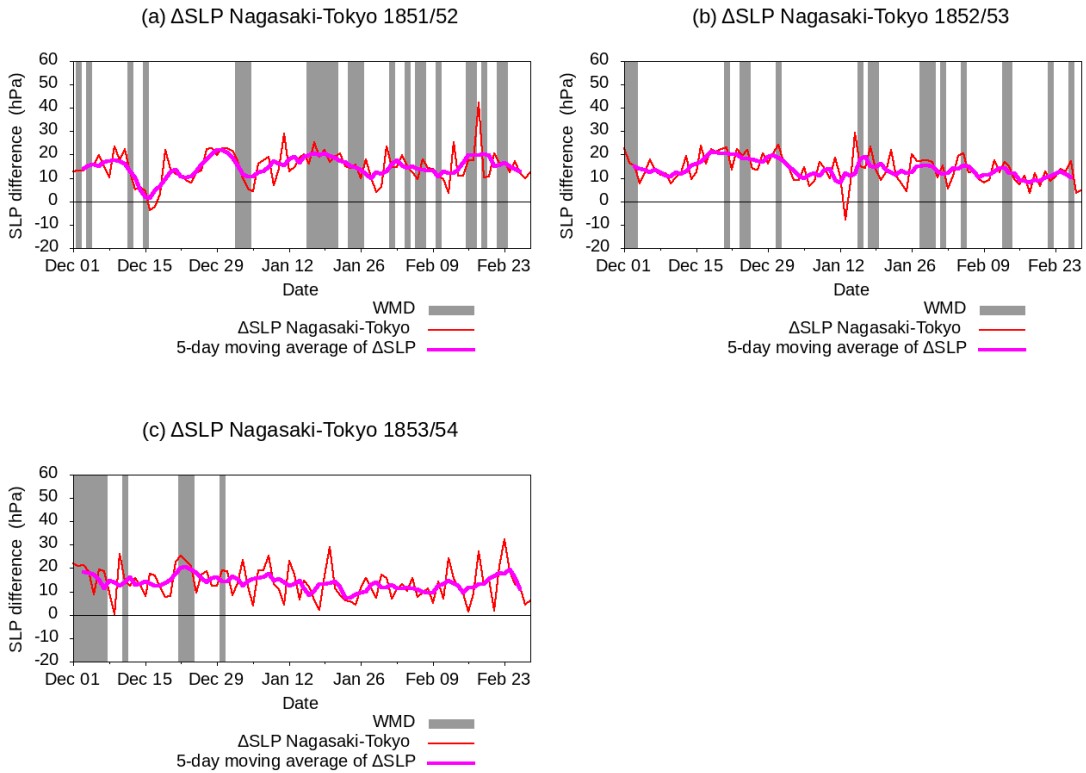

**Fig. 14 Temporal variations in surface pressure differences between Nagasaki and Tokyo (ΔSLP N-T) and winter monsoon outbreak dates (WMDs) for (a) 1851/52, (b) 1852/53, and (c) 1853/54.**

**Table 1: Comparison of weather descriptions from the Japan Meteorological Agency (JMA) Aomori observatory for 1–10 January 1980 and 1989**

|        | 1–10 January 1980          | 1–10 January 1989                        |
|--------|----------------------------|------------------------------------------|
| Jan 1  | Snow                       | Cloudy, rain, snow                       |
| Jan 2  | Fine                       | Cloudy, fine                             |
| Jan 3  | Cloudy                     | Cloudy, occasionally rain, snow          |
| Jan 4  | Cloudy, occasionally rain  | Snow, fine                               |
| Jan 5  | Snow                       | Snow                                     |
| Jan 6  | Cloudy, occasionally snow  | Cloudy, fine, occasionally snow          |
| Jan 7  | Snow                       | Fine                                     |
| Jan 8  | Snow                       | Cloudy, occasionally rain, fine          |
| Jan 9  | Snow                       | Slightly cloudy, occasionally rain, fine |
| Jan 10 | Snow                       | Fine, cloudy                             |