# Peer review of "Analysis of Early Japanese Meteorological Data and Historical Weather Documents to Reconstruct the Winter Climate between the 1840s and the Early 1850s"

_Climate of the Past, 2021_

## Referee Comment (RC1)

**Overall impression**

The paper examines the value of using historical weather records from diaries and pressure data to reconstruct past cold air disturbances (East Asian Winter Monsoon) over central Japan. Although this seems well-achieved to me, I would have liked to see a stronger outcome from this work. For instance, an historical case study is presented for 1851/2, yet in the conclusion it is argued that we still have much uncertainty about the apparent warm anomaly in the 1850s. So in that regard, it is disappointing that the paper, despite these efforts, is unable to shed much insight on past climate. I think it would very much strengthen the paper to temporally expand the 19th century record (analysis), which could then provide more substantive information on climate of the past – which I feel the current paper does not adequately achieve. For instance, it would be valuable to say something about longer-term changes (shifts) etc concerning the East Asian Winter Monsoon – and implications for climate over Japan…but this is currently not the case with the paper.

The methodological process seems robust enough to me and valuable. But as I argue for above, it really needs more done with it than only a look at 1851/2, and from which not too much is learnt.

**Moderate concerns**

The paper is reasonably well written but does require considerable editing with tightening up of text to meet the expectations of an international publication. In some places there are successive short sentences (each not overly informative). In such instances sentences could be combined to form slightly longer and more informative sentences, and thus provide for a better 'flowing' text. One example would be lines 112 to 115......too many short sentences...these could be combined and tightened up into a couple of sentences or so. But this is only an example – the entire paper would need to be carefully edited for this issue. I think a very thorough edit is required.

While I do not have a problem writing in the first-person plural – I find there is excessive use of the word 'We' in this manuscript ...in some cases 2 or more successive sentences using 'we' several times.

*Written tense:*

Too much of the paper is written in the past tense (which is not appropriate). I understand that in sections such as 'methodology' one writes in the past tense when referring to data that 'were collected' and specific analyses that 'were' undertaken etc. However, for the most part, the paper should be written in the present tense. Just a couple of examples to illustrate my point (but there are more than only these):

Lines 106/8: "*First, we investigated the temporal evolution of circulation fields and synoptic weather patterns for the present day (1968–1980). Subsequently, we investigated the East Asian winter monsoon activity....*" it would be better here (and elsewhere – where relevant [e.g. abstract etc]) to write in the present tense as you are currently investigating this through your publication presentation – even though the analysis in preparation for the paper is past tense. Hence I suggest write as: "*First, we*

*investigate the temporal evolution of circulation fields and synoptic weather patterns for the present day (1968–1980). Subsequently, we investigate the East Asian winter monsoon activity....*"

Another example from lines 120/1: "The other four locations (red circles in Fig. 1) were in the Pacific Ocean side, where dry weather prevailed." The four locations still exist today, so one cannot write as 'were' but should rather be written as 'are'. Dry weather prevailing on the leeward side is not something that happened only during past climates, but still happens today, hence one cannot write it as past (i.e. 'prevailed') tense. It should read as 'prevails'.

Figures:

Figure 1: Needs a scale bar.

Figure 2: This Figure (map) requires some quality improvement. Please shade terrestrial areas to differentiate from Oceanic areas. If not indicating elevation (as you do in Figure 1), then at least provide a grey scale to differentiate. Add names of Seas/Oceans. Needs a scale bar.

Figures 3 & 4. Need some indication of spatial context….so suggest adding some longitudinal/latitudinal values.

**Smaller technical items:**

**Title:** Instead of "Combined analysis of…." – I suggest rather say "Analysis of…….."

Line 39: "….an effective detection of outbreaks arising …." = a rather vague sentence. What type of outbreaks? Cold air outbreaks? Please specify.

Line 74: would read better as: "Historical Weather Data bank, based on information….."

Lines 78/9: "*Daily weather documents were documented simultaneously at various locations in Japan. Therefore, they are useful for reconstructing daily synoptic weather patterns*." This is all a bit vague. It is not clear if these 'weather documents' are something the paper aims to present or if reference is made to a pervious study that has documented these records ...and if so, what are these documents and who presented them?

Line 84: "….. in the Sea of Japan side." – would read better as: "...... on the windward (Sea of Japan) side of Japan." This may need to be addressed elsewhere in the paper where reference is made to 'Sea of Japan side'.

Lines 92/3: "Meanwhile, we recovered several early instrumental surface pressure series during the 19th century in Japan (Können et al., 2003; Zaiki et al., 2006, 2018)." This is again a bit vague here.........would be good to say more precisely for WHEN exactly (covering which years?)....and broadly for which area(s) of Japan? – is it central Honshu for instance?

Lines 94/6: "*Recently, we newly recovered surface air pressure observations in Beijing for the period 1841–1855, reported in "Annuaire 95 magnétique et métérologique du Corps des ingénieurs des mines de Russie" and "Annales de l'observatorie physique central https://doi.org/10.5194/cp-2021-33 Preprint. Discussion started: 12 April 2021 c Author(s) 2021. CC BY 4.0 License.4 de Russie" (Zaiki et al., 2008).*" **All this detail is not necessary as it can be obtained from the reference list. So rewrite as:** *"Recently, we newly recovered surface air pressure observations in Beijing for the period 1841–1855 (Zaiki et al., 2008)."*

Line 145: should be 'Dutch'

Line 180/1: should rather read as: "observations to compare with those in the…….."
Line 191: I would suggest rather write it as: "......selected 1 January 1868 to 31 December 1980 as the analysis period."

Lines 204/5: "three types: snowfall, rain, and fine or cloudy, according to the methodology of Yoshimura (2013)." Strictly speaking this is not correct as a fine (i.e clear sunny) day is not the same weather type as a 'cloudy' day. So it should really be four types, not three.

Sub-section title: **3.2 Analysis of sequence of circulation fields and weather pattern for the present day**
This seems a bit longwinded – needs a tightened up sub-section title. In fact, the titles of other sections and sub-sections could all do with some careful editing and tightening up.

---

## Author Response (AR1)

**Point by point reply to Editor**

We greatly appreciate the editor for carefully reading and commenting on our manuscript. In the following, we reply to his comments point by point. We marked up manuscript version showing the changes.

**Editor comment 1**

1. it would be very nice if you could give one or two short examples of the used "historical daily weather records" in chapter 2.1. Maybe in Japanese with English translation. The main purpose is to give readers who are not familiar with this kind of sources an idea.

**Author: reply to comment 1**

We greatly appreciate for this helpful suggestion. In the revised version of the article, we have added details about historical weather documents in line 85-88, p.3. We presented an example of daily weather records in the diary of the Hisrosaki Clan Agency Diaries in Fig.2. We translated Japanese weather description into English.

**Editor comment 2**

2. a few more details about the early surface pressure data. Is it known what instruments were used to measure them, how reliable are they, etc.? Here it is only about small additions, if information is known.

**Author: reply to comment 2**

Thank you for this helpful suggestion. Unfortunately, the information of type and accuracy of thermometers and barometers used during this period are very limited, the instruments were apparently brought from European countries as the data were recorded in western units such as Fahrenheit and English inch. The reliabilities of original data were carefully inspected by comparing modern meteorological station data, and the data which did not pass the inspections were eliminated. In the revised version of the article, we have added these explanations in line 114-118, p.4. We presented an example of temperature and pressure observations recorded in Calendar series (Reiken-koubo) as Fig.3.

**Editor comment 3**

3. line 200, p. 7 Centers instead of Canters

**Author: reply to comment 3**

Apologies for spelling the word incorrectly. We rewrite it as "Centers" in line 153, p. 6 in the revised manuscript.

**Point by point reply to reviewer1**

We greatly appreciate valuable comments and suggestions provided by reviewer #1. Below is a list of individual comments and questions followed by our replies:

**Reviewer1 comment1**

Overall Impression:

The paper examines the value of using historical weather records from diaries and pressure data to reconstruct past cold air disturbances (East Asian Winter Monsoon) over Central Japan. Although this seems well-archived to me, I would like to see a stronger outcome from this work. For instance, an historical case study is presented for 1851/52, yet in the conclusion it is argued that we still have much uncertainty about the apparent anomaly in the 1850s. So in that regard, it is disappointing that the paper, despite these efforts, is unable to shed much insight on past climate. I think it would very much strengthen the paper to temporally to expand the 19th century record(analysis), which could then provide more substantive information on climate of the past-which I feel the current paper does not adequately achieve. For instance, it would be valuable to say something about longer-term changes(shifts) etc concerning the East Asian Winter monsoon-and implications for climate over Japan ...but this is currently not the case with the paper.

The methodological process seems robust enough to me and valuable. But as I argue for above, it really needs more done with it than only a look at 1851/2, and from which not too much is learnt.

**Author: reply to comment 1**

Thank you for this helpful suggestion. We completely agree with this comment. In the revised manuscript, we analyzed interannual and intra-seasonal variations in the occurrence frequency of WMD for the period 1839/40 to 1853/54. For this analysis, we changed the methodology used to detect WMD, owing to the limited availability of pressure data and diary data sets. In the revised manuscript, we have detected WMD using two complete diary series for the windward (Sea of Japan) side of Japan and daily temperature data from Tokyo. Although we use only one temperature and two diary data sets, preliminary composite analysis (Fig.6-Fig.8) indicates that this method can be used to detect the East Asian winter monsoon outbreak. In addition, we compare reconstructed results with freezing dates records of Lake Suwa and extreme snowfall records reconstructed in China (Hao et al.,2011).

**Reviewer1 comment2:**

Moderate concerns:

This paper is reasonably well written but does require considerable editing with tightening up of text to meet the expectation of an international publication. In some places there are successive short sentences (each not overly informative). In such instances sentences could be combined to from slightly longer and more informative sentences, and thus providing for a better 'flowing' text. One example would be lines 112 to 115…too many short sentences. These could be combined and tightened up into a couple of sentences or so. But this is only an example-the entire paper would need to be carefully edited for this issue. I think very throughout edit is required.

While I do not have a problem writing in the first-person plural-I find there is excessive use of the word "We" in this manuscript...   in some cases, 2 or more successive sentences using 'we' several times.

**Author: reply to comment2**

Thank you for your valuable comment. We edited and revised the manuscript. In addition, the manuscript was checked by a native English speaker.

**Reviewer1: comment3**

Written tense:

Too much of the paper is written in the past sense (which is not appropriate). I understand that in section such as 'methodology' one writes in the past tense when referring to data that 'were collected' and specific analysis that 'were' undertaken etc. However, for the most part, the paper should be written in the present tense. Just a couple of examples to illustrate my point (but there are more than only these):

Lines 106/8: "First, we investigated the temporal evolution of circulation fields and synoptic weather patterns for the present day (1968–1980). Subsequently, we investigated the East Asian winter monsoon activity...." it would be better here (and elsewhere – where relevant [e.g. abstract etc]) to write in the present tense as you are currently investigating this through your publication presentation – even though the analysis in preparation for the paper is past tense. Hence I suggest write as: "First, we investigate the temporal evolution of circulation fields and synoptic weather patterns for the present day (1968–1980). Subsequently, we investigate the East Asian winter monsoon activity...."

Another example from lines 120/1: "The other four locations (red circles in Fig. 1) were in the Pacific

Ocean side, where dry weather prevailed." The four locations still exist today, so one cannot write as 'were' but should rather be written as 'are'. Dry weather prevailing on the leeward side is not something that happened only during past climates, but still happens today, hence one cannot write it as past (i.e. 'prevailed') tense. It should read as 'prevails'

**Author: reply to comment3:**

Thank you for noting this. We made the necessary changes in the revised manuscript regarding the use of the correct tense. Additionally, the manuscript was edited by a native English speaker.

**Reviewer1: comment4**

Figures

Figure 1:

Needs a scale bar.

**Author: reply to comment4**

Thank you very much. We have added a scale bar to Fig.1 of the revised manuscript.

**Reviewer1: comment5**

Figure 2:

This Figure (map) requires some quality improvement. Please shade terrestrial areas to differentiate from Oceanic areas. If not indicating elevation (as you do in Figure 1), then at least provide a grey scale to differentiate. Add names of Seas/Oceans. Needs a scale bar.

**Author: reply to comment5**

Thank you very much for your helpful comment. In the revised manuscript, we have indicated elevation in Fig.4 of the revised manuscript. In addition, we have added the names of the seas/oceans and a scale bar.

**Reviewer1: comment 6**

Figures 3 & 4:

Need some indication of spatial context….so suggest adding some longitudinal/ latitudinal

values.

**Author: reply to comment 6**

Thank you for this suggestion. In the revised manuscript, we have added longitudinal/latitudinal values to these figures (Fig.6-Fig.8 of the revised manuscript).

**Reviewer1: comment7**

Smaller technical items:

Title: Instead of "Combined analysis of…." – I suggest rather say "Analysis of……"

**Author: Reply to comment7**

Thank you for this suggestion. In the revised manuscript, we have modified the title to "Analysis of Early Japanese Meteorological Data and Historical Weather Documents to Reconstruct the Winter Climate between the 1840s and the Early 1850s".

**Reviewer1: comment8**

Line 39:

"….an effective detection of outbreaks arising …." = a rather vague sentence. What type of outbreaks? Cold air outbreaks? Please specify.

**Author: reply to comment8**

Thank you for this comment and apologies for the lack of clarity. In the revised manuscript, we rewrote this as "outbreak of the winter monsoon."

**Reviewer1: comment9**

Line 74:

would read better as: "Historical Weather Data bank, based on information…"

**Author: reply to comment9**

Thank you for this helpful comment. As suggested, we rewrote this as "Historical Weather Data Base, based on information…" in line39, p.2 of the revised manuscript.

**Reviewer1: comment10**

Lines 78/9:

"Daily weather documents were documented simultaneously at various locations in Japan. Therefore, they are useful for reconstructing daily synoptic weather patterns." This is all a bit vague. It is not clear if these 'weather documents' are something the paper aims to present or if reference is made to a pervious study that has documented these records ...and if so, what are these documents and who presented them?

**Author: reply to comment10**

Thank you for this suggestion. In the revised manuscript, we explain the use of data obtained from diaries in detail. This explanation includes descriptions of the documents and information on the authors of the documents. We addressed these explanations in line 85-line93, p.4 of the revised manuscript.

**Reviewer1: comment11**

Line 84:

"..... in the Sea of Japan side." – would read better as: "...... on the windward (Sea of Japan) side of Japan." This may need to be addressed elsewhere in the paper where reference is made to 'Sea of Japan side'

**Author: reply to comment 11**

Thank you for this suggestion. We follow your advice and rewrote this as "... on the windward (Sea of Japan) side of Japan."

**Reviewer1: comment12**

Lines 92/3:

"Meanwhile, we recovered several early instrumental surface pressure series during the 19th century in Japan (Können et al., 2003; Zaiki et al., 2006, 2018)." This is again a bit vague here.........would be good to say more precisely for WHEN exactly (covering which years?) ....and broadly for which area(s) of Japan? – is it central Honshu for instance

**Author: reply to comment12**

Thank you very much for this comment. My apologies for the confusion. In previous studies, early pressure and temperature data were recovered for Hakodate, Tokyo, Yokohama, Osaka, Kobe, and Nagasaki (Können et al., 2003; Zaiki et al., 2006). Except for Hakodate, most of these cities are in central and western Japan. Although there are several gaps, these early meteorological data cover the period from 1819. We added these explanations in line59 – line61, p.3 of the revised manuscript.

**Reviewer1: comment13**

Lines 94/6:

"Recently, we newly recovered surface air pressure observations in Beijing for the period 1841–1855, reported in "Annuaire 95 magnétique et métérologique du Corps des ingénieurs des mines de Russie" and "Annales de l'observatorie physique central https://doi.org/10.5194/cp-2021-33 Preprint. Discussion started: 12 April 2021 c Author(s) 2021. CC BY 4.0 License.4 de Russie" (Zaiki et al., 2008)." All this detail is not necessary as it can be obtained from the reference list. So rewrite as: "Recently, we newly recovered surface air pressure observations in Beijing for the period 1841–1855 (Zaiki et al., 2008)."

**Author: reply to comment13**

Thank you for this comment. We have deleted the unnecessary detail and rewrote following the reviewer's advice.

**Reviewer1: comment14**

Line 145:

should be 'Dutch'.

**Author: reply to comment14**

Apologies for spelling the word incorrectly. We rewrote it as" Dutch" in line108, p.4 of the revised manuscript.

**Reviewer1: comment15**

Line 180/1:

should rather read as: "observations to compare with those in the……."

**Author: reply to comment15**

Thank you for this suggestion. We rewrote this as "observations to compare with those in the…" as suggested by the reviewer. We rewrote it in line124 – line125, p.5 of the revised manuscript.

**Reviewer1: comment16**

Line 191:

I would suggest rather write it as: "......selected 1 January 1868 to 31 December 1980 as the analysis period."

**Author: reply to comment16**

Thank you for this comment. In the revised manuscript, we analyzed weather data from 1 December 1868 to 28 February 1980. We have deleted data for January 1968, February 1968, and December 1980, as there are no complete data sets available for the winter season (December–February) of these years. Therefore, we rewrote this section as "weather data for the years 1968/69–1979/80 were used to detect WMDs for the modern instrumental period". We rewrote this section in line130 – line131, p.5 of the revised manuscript.

**Reviewer1: comment17**

Lines 204/5:

"three types: snowfall, rain, and fine or cloudy, according to the methodology of Yoshimura (2013)." Strictly speaking this is not correct as a fine (i.e clear sunny) day is not the same weather type as a 'cloudy' day. So, it should really be four types, not three.

**Author: reply to comment17**

Thank you for pointing this out. We rewrote this as "four types." We rewrote this in line157, p.6 of the revised manuscript.

**Reviewer1: comment18**

Sub-section title: 3.2:

  Analysis of sequence of circulation fields and weather pattern for the present day

This seems a bit longwinded – needs a tightened up sub-section title. In fact, the titles of other sections and sub-sections could all do with some careful editing and tightening up.

**Author: reply to comment18**

Thank you for your comment. I agree with your suggestion. In the revised manuscript, we have changed the heading of this sub-section to "Analysis of circulation and precipitation patterns associated with WMDs" In addition, we rewrote the headings of other sections and sub-sections.

**Point by point reply to reviewer2**

We greatly appreciate valuable comments and suggestions provided by reviewer #2. Below is a list of individual comments and questions followed by our replies:

**Reviewer2: comment1**

In Fig.1, nine locations were selected to find out the synoptic weather patterns in Japan. Are they the only locations that have available historical weather records? If not, how are they selected.

**Author: reply to comment1**

  Many thanks for this valuable comment. Although historical weather documents are available for other locations, they often contain missing records or gaps. In particular, regarding the East Asian winter monsoon, very few continuous weather records are available for the area on the windward (Sea of Japan) side of Japan. Sparce coverage of complete weather records in this area could cause uncertainty in results, such as missed winter monsoon outbreaks.

  In the revised manuscript, we have analyzed interannual and intra-seasonal variations of the East Asian winter monsoon for the period from 1839/40 to 1853/54, following the reviewer's comments. For this analysis, it would be inappropriate to use sporadic weather records. Therefore, we selected two diary data sets (Hirosaki and Kawanishi, the first and second locations from north to south, Fig. 1) for the windward (Sea of Japan) side of Japan. Both diaries have almost complete daily weather records for the entire study period. With the exception of these two records, there are very few continuous diary data sets available for the Sea of Japan area. Furthermore, we used daily temperature series as observed in Tokyo to detect the outbreak of the East Asian winter monsoon. We addressed

**Reviewer2: comment2**

Also in Fig.1, the second location (counted from the north to the south) is considered as location in the Sea of Japan side. However, we can see from the elevation map that it locates in the northeast side of a mountain (or a high-altitude area), which is the leeward side relative to the East Asian winter monsoon. Also, according to Fig.5(d)-(e), it seems that there is limited precipitation in this location during WMDs. So, is it appropriate to classify it as the Sea of Japan side?

**Author: reply to comment2**

Thank you for this comment. According to the climatic divisions by Suzuki (1962), this location (Kawanishi, counted from the north to the south) falls within the Sea of Japan (windward) side climate zone. The climatic divisions by Suzuki (1962) are classified based on daily precipitation patterns in a typical "winter monsoon type pressure pattern day." Therefore, after careful consideration, we decide to retain the classification of this location as falling within the Sea of Japan side climate zone. In the revised manuscript, we refer to Suzuki (1962) concerning climatic divisions. We addressed these explanations to line88- line93, p.4 of the revised manuscript.

**Reviewer2: comment3**

According to line 182-183, the format of weather description in JMA data changed significantly after the mid-1980s, which makes it not appropriate for identification of synoptic weather patterns. Could you give an example in the text?

**Author: reply to comment3**

Thank you for this suggestion. In the revised manuscript, we presented an example of weather descriptions before and after the mid-1980s in Table1. In addition, we have addressed detailed explanation in line125- line130, p.5 of the revised manuscript.

**Reviewer2: comment4**

In line220, the continuous WMD days are not selected in the composited analysis (except the first WMD). But it's not quite clear to me that are there WMDs in the "preceding four days"? For example, suppose the WMD series from day 1 to day 5 is "WMD1-normal-WMD2-normal-WMD3",

are WMD2 and WMD3 adopted in the composite analysis, or only series like "normal-normal-normal-normal-WMD" adopted?

**Author: reply to comment4**

Apologies for the insufficient explanation in the original manuscript. We decide to use "normal-normal-normal-normal-WMD". We addressed these explanations to line 177 – line179,p.6 of the revised manuscript.

**Reviewer2: comment5**

In section 4.2, a case study is conducted for the winter of 1973/74. Why is this year selected? This seems a bit abrupt.

**Author: reply to comment5**

Thank you for this comment. In the revised manuscript, we have selected the coldest and warmest years based on seasonal (DJF) mean temperatures in Japan. From this, we conducted a case study for the selected winters. In the revised version of manuscript, relationship between SLP variations and WMDs for a cold winter year (1976/7) and those for a warm winter year (1978/9) are presented in Fig.9 and Fig.10.

**Reviewer2: comment6**

Similar to the last question, the historical case is for the winter of 1851/52. I think that the argument would be much more reliable if the study period is longer than one year. If the digitization of the surface pressure observation requires large amount of work, it is feasible to use the data of just Tokyo and Nagasaki, since this study already shows that SLP N-T is as effective as SLP B-T capturing the activity of the East Asian winter monsoon.

**Author: reply to comment6**

Thank you for pointing out this. In the revised manuscript, we have expanded the study period to extend from 1839/40 to 1853/54. Unfortunately, we found some quality problems in SLP data for Tokyo in the 1840s. Therefore, we decide not to use SLP data for the analysis of inter-annual variations during this period. However, daily temperature observations in Tokyo from 1839/40 to 1853/54 did not have any quality-related problems. Therefore, we detected "winter monsoon outbreak days (WMD)" by using daily temperature data for Tokyo and two diary data sets for the Sea of Japan side. In addition, we used SLP data only for a case study to confirm associations between the WMDs and

intra-seasonal variations of the east-west surface pressure gradient over East Asia (Fig.13-Fig.14).

Although we used a limited amount of data (daily temperature data for Tokyo and two diary data sets for the Sea of Japan side,) results from composite analysis (Fig.6-Fig.8) indicate that the outbreak of the East Asian winter monsoon can be detected from these data.

**Reviewer2: comment7**

In Fig.7(b) and (c), the bold purple line did not appear in the legend (also in Fig.8).

**Author: reply to comment7**

Thank you very much. Bold purple line indicates 5-day moving average. In the revised manuscript, we have added it in the legend.

---

## Author Response (AR2)

**Comments Editor**

Thank you very much for your revised paper. The new version is a very significant improvement and you have solved all major issues. There are only a few minor changes necessary, that are listed in report #1, and a last language check as proposed in report #2.

**Reply to Editor**

We greatly appreciate the editor for carefully reading and commenting on our manuscript. In the following, we reply to reviewer's comments point by point. We marked up manuscript version the changes comments indicated by red colors. In addition, we have checked English explanation again. We have changed several English explanations showing the changes by blue colors.

**Reviewer1 General comments:**

1. The methodology is neatly described. Analysis of "modern" meteorological data is used to describe the spatial patterns of precipitation and atmospheric pressure and temperature associated to the occurrences of winter monsoon outbreak days. These are used to identify the occurrence of these events during the middle XIX century.

**Author: reply to General comments**

We greatly appreciate for your comments, we have checked and corrected manuscript following your comments and suggestions.

**Reviewer1 Minor observations:**

Line 21-22: Change "was agreement" to "was in agreement"

**Author: reply to comment 1:**

Thank you for this helpful suggestion. In the revised manuscript, we have changed "was agreement" to "was in agreement" in line 22-23.

**Reviewer1 comment 2:**

Line 38: Yoshimura (2007) is not in the reference list.

**Author: reply to comment 2:**

Apologies for this mistake. We have added Yoshimura (2007) in the reference list. Line 424 of the revised manuscript.

**Reviewer1 comment 3:**

Line 63: Abdillah et al, 2021 appears as Abdillah et al, 2014 in the reference list.

**Author: reply to comment 3:**

Thank you for noting this. We have changed Abdillah et al, 2014 to Abdillah et al, 2021 in the reference list. Line 332 of the revised manuscript.

**Reviewer1 comment 4:**

Line 163: Change "the daily temperature anomaly in Tokyo was below its climatology" to "the daily temperature anomaly in Tokyo was negative"

**Author: reply to comment 4:**

Thank you very much for your helpful comment. In the revised manuscript, we have changed "the daily temperature anomaly in Tokyo was below its climatology" to "the daily temperature anomaly in Tokyo was negative" in line 164 of the revised manuscript.

**Reviewer1 comment 5:**

Line 193: Yamazaki et al., 2015 is not in the reference list.

**Author: reply to comment 5:**

Apologies for this mistake. We have added Yamazaki (2015) in the reference list. Line 420－421 of the revised manuscript.

**Reviewer1 comment 6:**

Line 200: Figure8. A space is missing before 8.

**Author: reply to comment 6:**

Thank you for noting this. We have added space before 8 in line 201 of the revised manuscript.

**Reviewer1 comment 7:**

Abdillah et al, 2021 appears as Abdillah et al, 2014 in the reference list

**Author: reply to comment 7:**

Thank you for noting this. We have changed Abdillah et al, 2014 to Abdillah et al, 2021 in the reference list. Line 332 of the revised manuscript.

**Reviewer1 comment 8:**

Lines 208 – 209: "Intra-seasonal variations in ΔSLP and WMDs for a cold winter year (1976/7) are presented in Fig. 9, and those for a warm winter year (1978/9) are presented in Fig. 10.". This is not what it is presented in Fig. 9 and Fig. 10. Each figure contains information for the winter years 1976/7 and 1978/9. Besides, figure captions of Fig. 9 and Fig. 10 are not consistent with the content of the two figures.

**Author: reply to comment 8:**

Thank you for this helpful suggestion. Apologies for incorrect explanations. In the revised manuscript, we have changed explanations in line 209－213. We also corrected figure captions of Fig. 9 and Fig. 10 of the revised manuscript.

**Reviewer1 comment 9:**

Line 343: Change "593. 1992." To "593, 1992."

**Author: reply to comment 9:**

Thank you for noting this. We have changed "593. 1992." To "593, 1992." in the reference list. Line 352 of the revised manuscript.

**Reviewer1 comment 10:**

Line 345: Add a space before 2011

**Author: reply to comment 10:**

Thank you for noting this. We have added a space before 2011 in the reference list. Line 354 of the revised manuscript.

**Reviewer1 comment 11:**

Line 347: Add a space before 2008

**Author: reply to comment 11:**

Thank you for noting this. We have added a space before 2008 in the reference list. Line 356 of the revised manuscript.

**Reviewer1 comment 12:**

Line 387: Add a space before 2009

**Author: reply to comment 12:**

Thank you for noting this. We have added a space before 2009 in the reference list. Line 397 of the revised manuscript.

**Reviewer1 comment 13:**

Line 390: Add a space before 2021.

**Author: reply to comment 13:**

Thank you for noting this. We have added a space before 2021 in the reference list. Line 399 of the revised manuscript.

**Reviewer1 comment 14:**

Line 406: Add a space before 2015.

**Author: reply to comment 14:**

Thank you for noting this. We have added a space before 2015 in the reference list. Line 415 of the revised manuscript.

**Reviewer1 comment 15:**

Line 414: Change "68. 2013" to "68, 2013"

**Author: reply to comment 15:**

Thank you for noting this. We have changed "68. 2013" to "68, 2013" in line 426 of the revised manuscript.

**Reviewer1 comment 16:**

Lines 518 – 520: Indicate in the figure caption the period used to calculate the temperature anomalies.

**Author: reply to comment 16:**

Thank you for this helpful suggestion. We have indicated the period used to calculate the temperature anomalies in line 532 of the revised manuscript.

**Reviewer1 comment 17:**

Line535: I suggest in each panel to indicate by dot or a circle the position of Japan.

**Author: reply to comment 17:**

Thank you for this helpful suggestion. We have indicated the position of Japan by a circle in Fig.6 of the revised manuscript.

**Reviewer1 comment 18:**

Line548: I suggest in each panel to indicate by dot or a circle the position of Japan.

**Author: reply to comment 18:**

Thank you for this helpful suggestion. We have indicated the position of Japan by a circle in Fig.7 of the revised manuscript.

**Reviewer2 General comments:**

The paper has great potential to be published in the journal. According to the current version, the authors have done lots of effort to revise the manuscript according to the comments from reviewers. The authors have provided a good reason on selecting location for their analysis considering both climate system and records condition. Not only the location, the selection of study period and comparison with 1976-1977 has been explained well in this revision. I am also glad to see the vivid examples of Japanese records, which will provide the audience more information. From these examples, the audience should be more familiar with the Japanese records for studying past climate.

Overall, the revision is satisfied to me. The language is also acceptable, but still has room to be improved. To make it more readable, I suggest the author to make a list for abbreviation because there are so many abbreviations in the paper.

**Author: reply to reviewer 2:**

We greatly appreciate valuable comments and suggestions. We have checked language of manuscript again and made a list for abbreviation in line $303-309$ of the revised manuscript.